# Reliability and validity study of the Spanish adaptation of the "Educational Practices Questionnaire" (EPQ)

**Mariona Farrés-Tarafa**[1,2,3,4], **Juan Roldán-Merino**[1,2,5,6]*, **Urbano Lorenzo-Seva**[7], **Barbara Hurtado-Pardos**[1,2,8], **Ainoa Biurrun-Garrido**[1,2], **Lorena Molina-Raya**[1,2], **Maria-Jose Morera-Pomarede**[1,2], **David Bande**[9], **Marta Raurell-Torredà**[10,11,12], **Irma Casas**[13,14,15]

**1** Campus Docent, Sant Joan de Déu—Fundació Privada, School of Nursing, University of Barcelona, Barcelona, Spain, **2** Research Group GIES (Grupo de investigación en Enfermería, Educación y Sociedad), Barcelona, Spain, **3** Member Research Group GRISIMula (Grupo emergente 2017 SGR 531; Grupo en Recerca Enfermera en Simulación), Barcelona, Spain, **4** Secretaria Research Group GRISCA (Grupo en Recerca Enfermera en Simulación en Cataluña y Andorra), Barcelona, Spain, **5** Research Group GEIMAC (Consolidated Group 2017–1681: Group of Studies of Invarianza of the Instruments of Measurement and Analysis of Change in the Social and Health Areas), Barcelona, Spain, **6** Coordinator Research Group GIRISAME (International Researchers Group of Mental Health Nursing Care), Barcelona, Spain, **7** Universitat Rovira I Virgili, Tarragona, Spain, **8** Member Research Group GRIN (Grupo Consolidado de Recerca Infermeria, SRG:664), Barcelona, Spain, **9** Anesthesiologist, Servicio Anestesiología, Reanimación y Tratamiento del Dolor, Parc de Salut Mar, Barcelona, Spain, **10** Universidad de Barcelona, Barcelona, Spain, **11** Presidenta Sociedad Española de Enfermería Intensiva y Unidades Coronarias (SEEIUC), Madrid, Spain, **12** President Research Group GRISIMula (Grupo emergente 2017 SGR 531; Grupo en Recerca Enfermera en Simulación), Barcelona, Spain, **13** Universitat Autònoma de Barcelona, Barcelona, Spain, **14** Preventive Medicine Service, Hospital Germans Trias i Pujol, Barcelona, Spain, **15** Research Group Innovation in Respiratory Infections and Tuberculosis Diagnosis (Group Consolidat 2017 SGR 494)

* jroldan@santjoandedeu.edu.es

**Data Availability Statement:** All relevant data are within the manuscript and its Supporting Information files.

## Abstract

The Educational Practices Questionnaire is an instrument for assessing students perceptions of best educational practices in simulation. As for other countries, in Spain, it is necessary to have validated rubrics to measure the effects of simulation. The objective of this study was to carry out a translation and cultural adaptation of the Educational Practices Questionnaire into Spanish and analyze its reliability and validity. The study was carried out in two phases: (1) adaptation of the questionnaire into Spanish. (2) Cross-sectional study in a sample of 626 nursing students. Psychometric properties were analyzed in terms of reliability and construct validity by confirmatory and exploratory factor analysis. The exploratory and confirmatory factor analyses showed that the one-dimensional model is acceptable for both scales (presence and importance). The results show that the participants' scores can be calculated and interpreted for the general factor and also for the four subscales. Cronbach's alpha and the Omega Index were also suitable for all the scales and for each of the dimensions. The Educational Practices Questionnaire is a simple and easy-to-administer tool to measure how nursing degree students perceive the presence and importance of best educational practices.

**Funding:** The author(s) received no specific funding for this work.

**Competing interests:** The authors have declared that no competing interests exist.

## Introduction

Using clinical simulation as a tool for teaching both new nursing professionals during their university education and existing professionals during continuing education has grown exponentially in recent years [1]. Simulation is also regarded as an effective educational method for the delivery of clinical scenarios [2].

The literature states that, in order to obtain optimal learning results through simulation associated with the competencies that nurses must master in their clinical practice, it is necessary to use a common international language and these activities must be incorporated throughout the entire Nursing Degree curriculum [3]. In addition, the International Nursing Association for Clinical Simulation and Learning (INACLS), and several studies [4, 5] affirm that it is necessary to establish best practices for the simulation methodology to be effective. In 2005, Pamela Jeffries developed a guide on simulation methodology for nursing education through the *National League for Nursing (NLN)* along with *Laerdal*; this consisted of 5 basic components necessary to conduct a simulation session: educational practices, the facilitator, participants, simulation design features and expected results [6].

Based on the seven principles of good practice defined by Chickering & Gamson, educational practices related to simulation were defined [7]. These practices are: 1) Active learning: through simulation, students actively learn, as they have the opportunity to participate directly in the activity, both when performing the scenario and in subsequent debriefing [8]; 2) Feedback: simulation offers immediate feedback, from the instructor and classmates, as well as from the human patient simulator (HPS), on the knowledge and skills demonstrated and the decisions made [9]; 3) Interaction: intercommunication between the university and the student fosters a climate of trust between the instructor and the students. Together they can discuss and reflect on the learning process, in addition to designing individualized improvement action plans according to the needs of each one.[10]; 4) Collaborative learning: simulation promotes collaborative learning as it provides a reality-like environment where all participants work together for the same purpose and share the decision-making process [11]. This can offer several advantages, allowing participants to learn from different disciplines and learn about teamwork and, if they have different levels, novice nurses can even be given the opportunity to learn from experts [12]; 5) High expectations: it is important that the expectations before performing the simulation are high, so that both students and instructors feel empowered to achieve greater learning in a safe environment [13]; 6) Diversity in learning: people have different learning needs depending on their personal characteristics. It is important to implement different teaching methodologies in the curricula, including simulation [14] and 7) Time: through simulation we can give training in techniques to reduce times in real clinical practice [15].

Subsequently, in order to understand how educational practices on high fidelity simulation are perceived by participants, the *NLN*, along with Laerdal, developed the *"Educational Practices Questionnaire" (EPQ)* to evaluate perceptions of best educational practices in simulation [16].

In the Spanish educational context there are several valid and reliable tools that can be used to know the level of satisfaction of students about new teaching methodologies; these include the tutorial action [17, 18], or simulation plans [19, 20]. There are also tools to measure nursing competences in a simulation scenario [21–23], to evaluate the debriefing through the DASH report [24]. However, in Spain, as occurs in other countries, there are no validated instruments that evaluate perceptions of best educational practices in simulation [25]. Therefore, it is essential to have validated rubrics in Spain in order to be able to evaluate the effects of simulation activities.

The objective of this study was to carry out a translation and cultural adaptation of the *Educational Practices Questionnaire* into Spanish and analyze its reliability and validity.

## Methods

### Design

A two-phase study was conducted. In the first phase, the instrument was adapted to Spanish; and in the second phase, the metric properties of the EPQ questionnaire translated into Spanish were analyzed.

### Participants and setting

The study sample consisted of 626 nursing students from the 2018–19 academic year at the Campus Docent Sant Joan de Déu Fundació Privada [Sant Joan de Déu Private Foundation Teaching Campus], a center affiliated with the Universidad de Barcelona [University of Barcelona]. Non-probability convenience sampling was used. Students who had performed a clinical simulation during the course were included and only those who were not present at the time of the simulation were excluded.

To calculate the sample size, the recommendations of Comrey and Lee [26] for validation studies were followed, which consider a good sample to be anything more than 500 participants.

### Variables and source of information

All items related to the EPQ questionnaire were collected as variables. It is a questionnaire made up of 16 items that are grouped into four dimensions (active learning, collaboration, different ways of learning, and expectations). For each item, the same questionnaire makes it possible to assess both the presence of best educational practices and the importance of best practices integrated in clinical simulation [16].

Each item is evaluated using a scale with five possible answers, where 1) is strongly disagree, 2) disagree, 3) undecided, 4) agree, and 5) strongly agree. The sum of the scores of all the items represents a greater recognition of best educational practices in simulation.

In the study conducted by Jeffries and Rizzolo [16], 'the presence of educational best practices' obtained a Cronbach's alpha of 0.86 and 'importance of best practices embedded in simulation' obtained 0.91. This reliability was similar to what was found in another study, which was 0.95 [27].

Other variables were also collected such as: age, sex, teaching shift, whether they were working, work shift and whether they had previous work experience in the health field.

### Procedure

The study was conducted in two phases, the first of which consisted of translating and adapting the English version into Spanish through an independent bilingual English—Spanish, Spanish—English committee.

Fig 1 shows the entire translation and back-translation process that was performed following the Standards for Educational and Psychological Testing [28]. Table 1 shows the semantic equivalence of items from English to Spanish.

Finally, a pilot test was done, and the participants (n = 15) concluded that it required little time for completion (5 to 10 minutes) and that it was easy to understand. At second phase, the questionnaire was administered to the nursing students who had performed a simulation to analyze the psychometric properties of the Spanish version. The Spanish version was named *EPQ-Sp*.

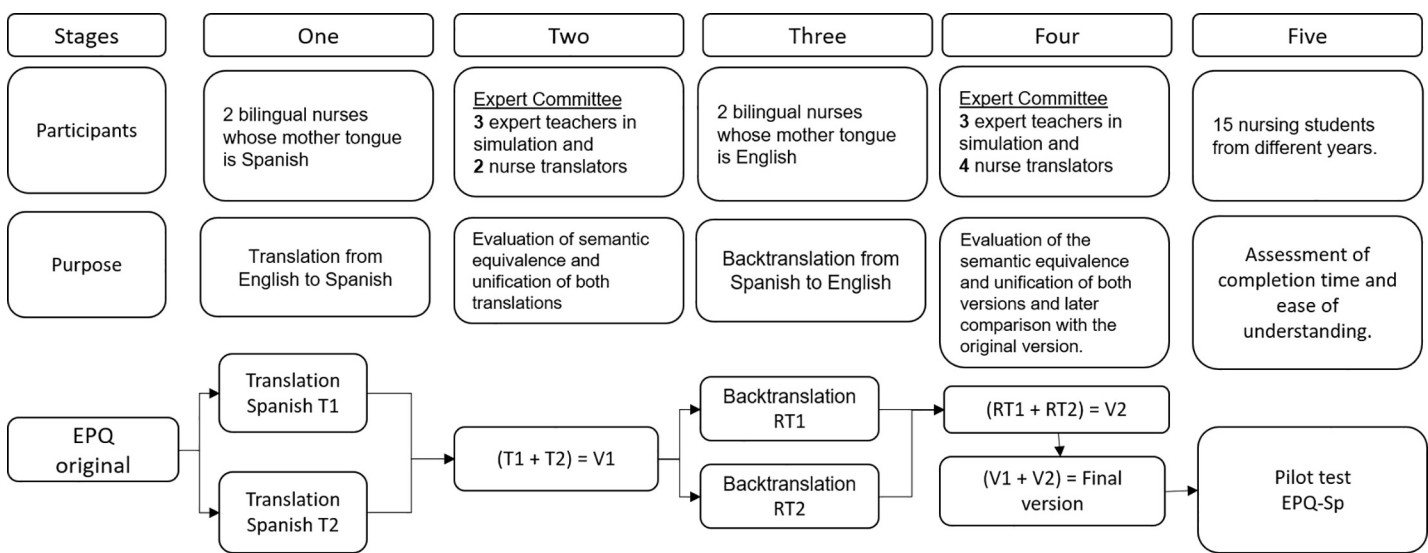

**T1**. First Translation Spanish; **T2** Second translation Spanish; **V1**. First version English ; **V2**. Second version English; **RT1**;  First Backtranslation; **RT2**. Second Backtranslation

**Fig 1.  Description of phase: Adaptation to Spanish of the "Educational Practices Questionnaire" (EPQ).**

## Statistical analysis

Confirmatory factorial analysis (CFA) models were estimated using structural equation modelling on the polychoric correlation matrix (EQS 6.2 for Windows, Multivariate Software, Inc., Encino, CA, USA). It is important to note that *Presence* and *Importance in Simulation* scales were analyzed as separated scales (i.e., factorial analyses were computed first for the 16 items related to *Presence scale*, and then for the 16 items related to the *Importance in Simulation scale*).

A CFA was performed to analyze the validity of the construct using the generalized least squares method. The goodness of the fit was examined in terms of the standardized Chi-square, defined as the ratio between the value of the Chi-square and the number of degrees of freedom ($\chi 2/df$), Adjusted Goodness of Fit Index (AGFI), Goodness of Fit Index (GFI), Comparative Fit Index (CFI), Bentler Bonnet Non-Normed Fit Index (BBNNFI), Bentler Bonnet Normed Fit Index (BNNFI), Root Mean Standard Error Standardized (RMRS) and Root Mean Standard Error of Approximation (RMSEA). A good overall adjustment was considered if the adjustment values were: $X^2/df$ between 2 and 6 [29]; AGFI, GFI, CFI, BBNNFI, BBNFI $\geq$ .90 and RMRS, RMSEA $\leq$ .06 [30–32]. Reliability was analyzed using Cronbach's alpha coefficient [33] as well as the Omega Index [34] since this latter index makes it possible to analyze the degree of internal consistency based on the factorial loads and does not depend on the number of items, as the alpha coefficient does, which is appropriate given that three of the four dimensions (D2. Collaboration, D3. Diverse ways of learning and D4. High expectations) have two items.

The following were considered acceptable values: Cronbach's alpha values greater than 0.70 [33, 35] and Omega Index values greater than 0.80 [34].

The inspection of the estimated parameters of the CFA suggested that a unidimensional factor solution could also be a plausible option. In order to assess whether the scale could be considered as essentially unidimensional, we computed Explained Common Variance (ECV) and Unidimensional Congruence (UniCo) indices to assess the degree of dominance of the general

**Table 1. Shows the semantic equivalence of items from English to Spanish that were metrically validated.**

| | English | Spanish |
|---|---|---|
| **Item 1** | I had the opportunity during the simulation activity to discuss the ideas and concepts taught in the course with the teacher and other students. | Durante la actividad de simulación tuve la oportunidad de debatir sobre ideas y conceptos presentados con el instructor/facilitador y el resto de los estudiantes. |
| **Item 2** | I actively participated in the debriefing session after the simulation. | Participé activamente en el debriefing posterior a la simulación. |
| **Item 3** | I had the opportunity to put more thought into my comments during the debriefing session. | Durante el debriefing tuve la oportunidad de reflexionar más sobre mis comentarios. |
| **Item 4** | There were enough opportunities in the simulation to find out if I clearly understand the material. | Durante la simulación hubo suficientes oportunidades de saber si entendía bien el material. |
| **Item 5** | I learned from the comments made by the teacher before, during, or after the simulation. | Aprendí de los comentarios del instructor/facilitador, antes, durante o después de la simulación. |
| **Item 6** | I received cues during the simulation in a timely manner. | A lo largo de la simulación recibí indicaciones puntuales. |
| **Item 7** | I had the chance to discuss the simulation objectives with my teacher. | Tuve la oportunidad de hablar de los objetivos de la simulación con el instructor/facilitador. |
| **Item 8** | I had the opportunity to discuss ideas and concepts taught in the simulation with my instructor. | Tuve la oportunidad de debatir sobre ideas y conceptos presentados en la simulación con el instructor/facilitador. |
| **Item 9** | The instructor was able to respond to the individual needs of learners during the simulation. | El instructor/facilitador pudo responder a las necesidades individuales de los estudiantes durante la simulación. |
| **Item 10** | Using simulation activities made my learning time more productive. | Gracias a las actividades de simulación, mi tiempo de aprendizaje fue más productivo. |
| **Item 11** | I had the chance to work with my peers during the simulation. | Tuve la oportunidad de trabajar con mis compañeros durante la simulación. |
| **Item 12** | During the simulation, my peers and I had to work on the clinical situation together. | Durante la simulación, mis compañeros y yo tuvimos que trabajar juntos en la situación clínica. |
| **Item 13** | The simulation offered a variety of ways in which to learn the material. | La simulación ofreció varias maneras de aprender el material. |
| **Item 14** | This simulation offered a variety ways of assessing my learning. | La simulación ofreció varias maneras de valorar el aprendizaje. |
| **Item 15** | The objectives for the simulation experience were clear and easy to understand. | Los objetivos de la experiencia de simulación eran claros y fáciles de comprender. |
| **Item 16** | My instructor communicated the goals and expectations to accomplish during the simulation | El instructor/facilitador comunicó los objetivos y expectativas que había que alcanzar durante la simulación. |

factor or closeness to unidimensionality [36]. The ECV index essentially measures the proportion of common variance of the item scores that can be accounted for by the first canonical factor (i.e. the factor that explains most common variance). The UniCo index is the congruence between the actual loading matrix and the loading matrix that would be obtained if the unidimensional model is true: the closer to the value of 1, the more the actual loading matrix looks like the unidimensional loading matrix. To conclude that a scale is essentially unidimensional, ECV and UniCo values should be larger than 0.850 for ECV [36], and 0.950 for UniCo [37]. Finally, we also computed Optimal Implementation of Parallel Analysis (PA) [38].

In order to explore the loading values of the items in a unidimensional solution, an exploratory factor analysis (EFA) was computed. Item scores were treated as ordered-categorical variables and the EFA was fitted to the inter-item polychoric correlation matrix [39]. The chosen fitting function was robust unweighted least squares, with mean-and-variance corrected fit statistics [40]. A single factor was extracted.

We were also interested in assessing a bifactor model for the *Presence* scale. Factor analysis applications to item are generally based on one of these two models: (a) the unidimensional (Spearman) model or (b) the correlated-factors model. The bifactor model combines both previous models: it allows the hypothesis of a general dimension to be maintained, while the additional common variance among the scores is modelled using group factors that are expected to approach a simple structure [41]. In particular, we computed Pure Exploratory Bifactor (Pebi) proposed by Lorenzo-Seva & Ferrando and implemented in Factor software [42]. As the scale was aimed at measuring four factors that were expected to approach a simple structure, these factors were rotated using Robust Promin rotation [42].

### Ethical considerations

The study was approved by the Clinical Investigation Ethics Committee of the Sant Joan de Déu Foundation with CEIC research code PIC-42-19. All participants were informed of the purpose of the study and they freely gave their verbal and written consent to participate in the study as volunteers. The translation has been completed with the consent of the National League for Nursing (NLN), but NLN is not responsible for its accuracy. NLN holds the copyright to the original (English language) and the translated instrument in Spanish. Any request related to the translated instrument in Spanish must be addressed to NLN. More information about research instruments and copyright is available in NLN website [http://www.nln.org/professional-development-programs/research/tools-and-instruments]

## Results

### Demographic characteristics

Finally, a total of 626 nursing students were included in the study. The mean age was 22.9 (SD 5.1), 83.4% were women. More than half of the students (57.7%) were enrolled in the morning study schedule. 74.4% of the students declared that they were working at that time and of these 62.4% had temporary employment (Table 2).

### Construct validity

In the following subsections, we present the different analyses that we computed to assess construct validity: Confirmatory Factor Analysis (CFA), Essential Unidimensionality, and Exploratory Bifactor (PEBI).

**Table 2. Sociodemographic characteristics of the study population.**

|  | n | % |
|---|---|---|
| Age (SD) | 22.9 (SD 5.1) | |
| Sex | | |
| Women | 522 | 83.4 |
| Men | 104 | 16.6 |
| Study schedule. | | |
| Morning | 361 | 57.7 |
| Afternoon | 265 | 42.3 |
| Currently employed | | |
| Yes | 466 | 74.4 |
| Not | 160 | 25.6 |
| Type of contract | | |
| Permanent employment | 175 | 37.6 |
| Temporary employment | 291 | 62.4 |

**Confirmatory Factor Analysis (CFA).** The confirmatory factorial analysis was used to verify the internal structure of the questionnaire, in which a 4-dimensional model identical to the structure of the original version of the questionnaire was proposed. Parameter estimation was performed using the least squares method. This method is usually used primarily for ordinal measurement items and has the same properties as the maximum likelihood method, although under less stringent multivariate normality considerations [43].

Dimensions 2, 3 and 4 have the highest factor loads or saturations for both assessments (presence and importance of simulation). All saturations were greater than 0.50. The correlations between the factors for the presence and importance of the simulation were high (Figs 2 and 3, respectively).

The Chi square test was statistically significant but the fit ratio was 4.17 (presence of good practices) and 5.32 (importance), so if it is between 2–6 the fit is reasonably good [43]. Likewise, the rest of the indices analyzed present the same trend, so it can be concluded that the model correctly fits (Table 3).

**Essential unidimensionality.** As can be observed in Figs 2 and 3, the CFA model was adjusted for a model where the factors were strongly correlated between themselves. This outcome was observed for both the *Presence* and *Importance in Simulation* scales. As the correlations were large, a unidimensional factor model could also be expected to fit properly. To evaluate this hypothesis, we computed an analysis to assess essential unidimensionality. For the *Presence* scale, the ECV and UniCo values were 0.845 and 0.973, respectively. For the *Importance in Simulation* scale, the ECV and UniCo values were 0.849 and 0.979, respectively. The values of both indices suggested that there is a dominant factor running through all the 16 items in both scales. In addition, the first eigenvalue of the *Presence* and *Importance in Simulation* scales accounted for 50.9% and 58.2% of the common variance, respectively. PA suggested that the unidimensional solution is the most replicable in both cases.

Goodness of fit indices for the single factor model are printed in Table 4. As can be observed in the table, the fit is not so good as the multidimensional model tested in CFA, but it is still acceptable. Finally, Expected at Posteriori reliability [44] of the single factor was 0.926 and 0.947 for the *Presence* and *Importance in Simulation* scales, respectively.

**Exploratory bifactor (PEBI).** While the unidimensional solution is acceptable for both the *Presence* and *Importance in Simulation* scales, the fit indices inspected in the previous subsection show that the fit to the unidimensional solution is slightly worse in the case of the *Presence* scale. This means that, while the general factor for *Presence* is strong, the four group factors can still play a substantial role in the factor model. To test this hypothesis, we fitted an exploratory bifactor model related to the *Presence* scale. Goodness of fit indices for the bifactor model are printed in Table 5. As can be observed in the table, the fit was actually very good. In addition, all items had a salient loading in the general factor and in the expected group factor (see the loading matrix printed in Table 6). It must be pointed out that some items expected to measure *Active learning* (i.e., items 4, 5, 7 and 9) also loaded in *High expectation*. The correlation among group factors ranged from 0.04 to 0.20. Finally, Orion reliabilities of factors [44] ranged from 0.712 (for *High expectation)* to 0.920 (for *Active learning*). The general factor showed an Orion reliability of 0.881.

**Conclusion of construct validity analyses.** The conclusion is that both models (unidimensional and multidimensional) are acceptable. From a practical point of view, this means that researchers can compute the overall scale score (i.e., the score that is obtained using the responses to the all items), but also the score in four subscales when a more detailed description of participant responses may be needed.

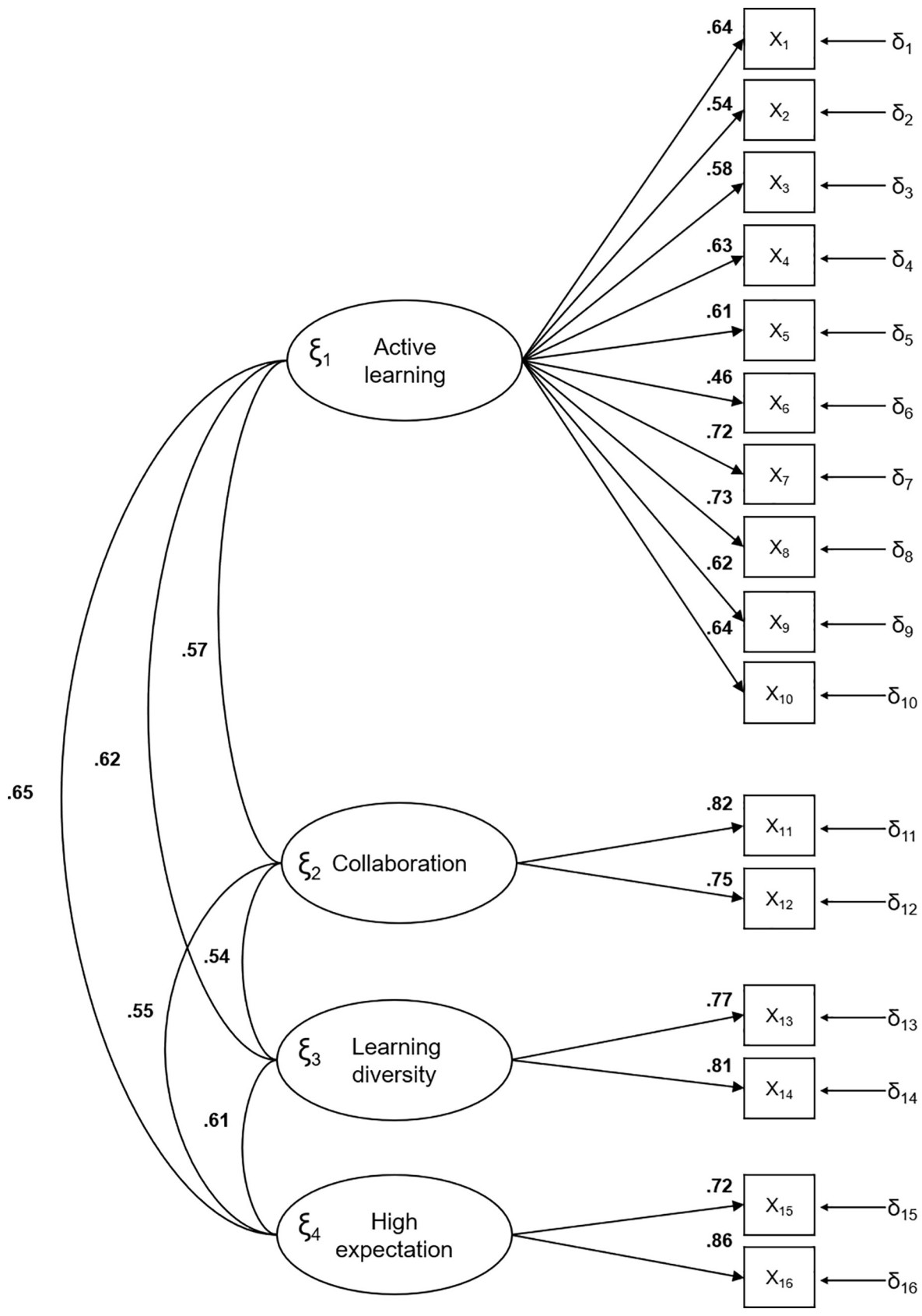

**Fig 2. Standardized model parameters for the presence of educational best practices.**

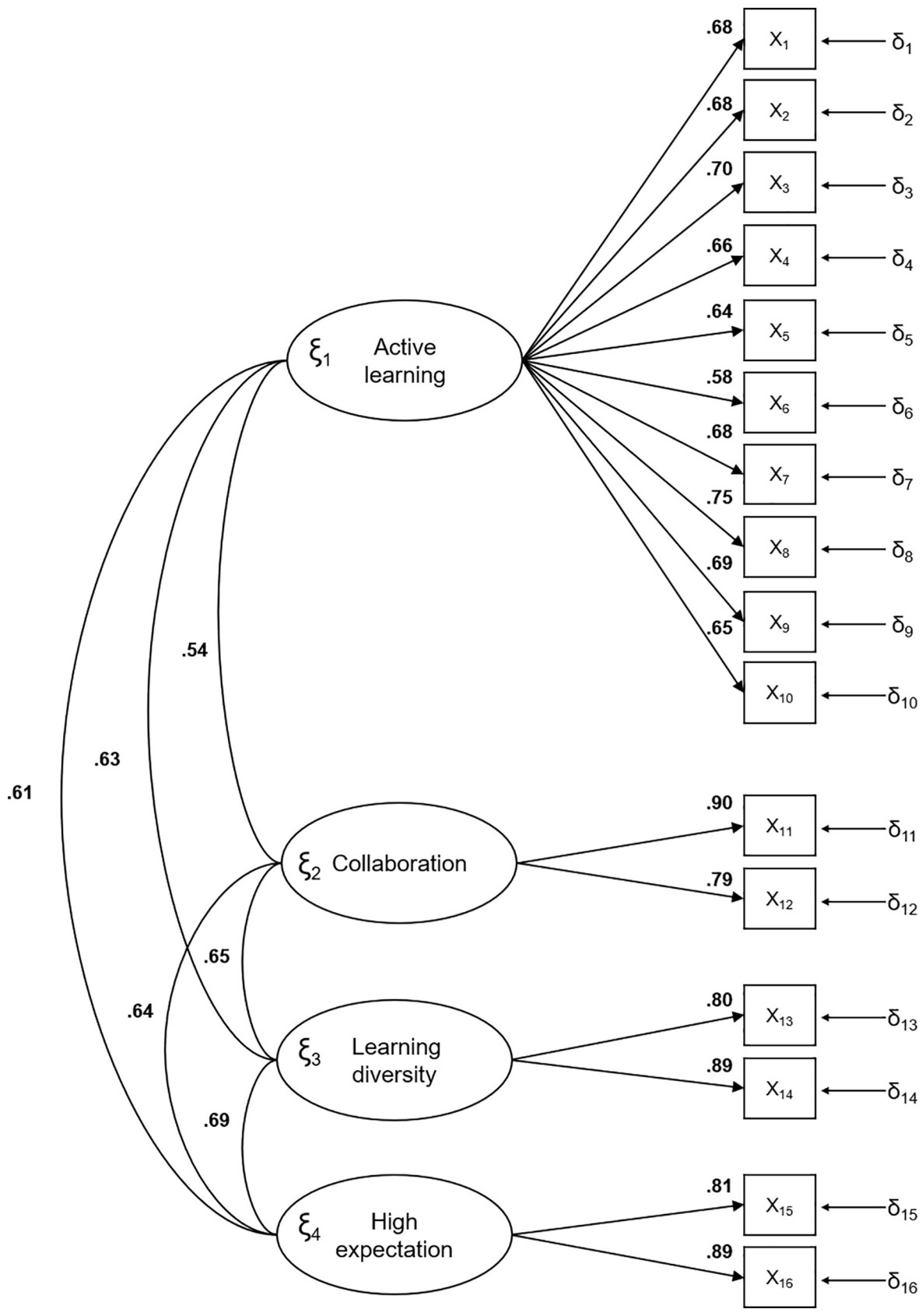

**Fig 3. Standardized model parameters for the importance of educational best practices.**

**Table 3. Indices of goodness of fit of the confirmatory model.**

| INDEX | Assess perceptions of educational best practices | |
| --- | --- | --- |
| | **Presence** | **Importance in simulation** |
| | **VALUE** | **VALUE** |
| BBNFI | .895 | .899 |
| BBNNFI | .900 | .897 |
| CFI | .918 | .916 |
| GFI | .918 | .900 |
| AGFI | .886 | .861 |
| RMSR | .047 | .046 |
| RMSEA | .071 (90% CI: .064 -.078) | .083 (90% CI: .076 - .090) |
| α Cronbach | .894 | .915 |
| Goodness of fit test | $\chi^2$ = 408.723; gl = 98; $P$ < 0.0001 | $\chi^2$ = 522,125; gl = 98; $P$ < 0.0001 |
| Reason for fit | $\chi^2$ / gl = 4.17 between 2–6 | $\chi^2$ / gl = 5.32 between 2–6 |

BBNFI: Bentler Bonnet Normed Fit Index. BBNNFI: Bentler Bonnet Non-Normed Fit Index CFI: Comparative Fit Index. GFI: Goodness of Fit Index. AGFI: Adjusted Goodness of Fit Index. RMSR: Root Mean Standard Error Standardized. RMSEA: Root Mean Standard Error of Approximation. CI Confidence Interval

## Reliability

Cronbach's alpha internal consistency coefficient for the total of the Assess perceptions of educational best practices presence and importance and simulation questionnaire was 0.894 and 0.915, respectively. The Omega (ω) coefficient for the questionnaire total was 0.922 (presence) and 0.945 (Importance and simulation). All values obtained for each dimension and each coefficient were greater than 0.762 (Table 7).

## Discussion

This study describes the adaptation to Spanish and the psychometric analysis of the "Educational Practices Questionnaire" (EPQ). It is a questionnaire made up of 16 items designed to evaluate both the presence of best educational practices and the importance of best practices integrated in simulation. The results show that the Spanish EPQ has adequate psychometric properties in terms of internal consistency and the validity of the construct. Internal consistency calculated with Cronbach's alpha coefficient was adequate ($\alpha \geq 0.70$) for the total of the

**Table 4. Indices of goodness of fit of the exploratory unidimension to the model.**

| INDEX | Assess perceptions of educational best practices' | | | |
| --- | --- | --- | --- | --- |
| | **Presence** | | **Importance in simulation** | |
| | **VALUE** | **95% CI** | **VALUE** | **95% CI** |
| CFI | .968 | (.958 - .982) | .973 | (.962 - .983) |
| GFI | .970 | (.963 - .981) | .973 | (.964 - .983) |
| AGFI | .966 | (.957 - .978) | .969 | (.957 - .978) |
| RMSEA | .083 | (.069 - .089) | .094 | (.078 - .105) |
| Goodness of fit test | $\chi^2$ = 554.578; gl = 104; $P$ < .0001 | | $\chi^2$ = 680.538; gl = 104; $P$ < .0001 | |
| Reason for fit | $\chi^2$ / gl = 5.3 between 2–6 | | $\chi^2$ / gl = 5.8 between 2–6 | |

CFI: Comparative Fit Index. GFI: Goodness of Fit Index. AGFI: Adjusted Goodness of Fit Index.
RMSEA: Root Mean Standard Error of Approximation CI: Confidence Interval

**Table 5. Indices of goodness of fit of the exploratory bifactor model of *Presence* scale.**

| INDEX | Assess perceptions of educational best practices | |
|---|---|---|
| | Presence | |
| | VALUE | 95% CI |
| CFI | .998 | (.998 - .999) |
| GFI | .998 | (.998 - .999) |
| AGFI | .995 | (.995 - .997) |
| RMSEA | .029 | (.012 - .029) |
| Goodness of fit test | $\chi^2 = 36.514$; gl = 50; $P < .0001$ | |
| Reason for fit | $\chi^2$ / gl = 5.3 between 2–6 | |

CFI: Comparative Fit Index. GFI: Goodness of Fit Index. AGFI: Adjusted Goodness of Fit Index.
RMSEA: Root Mean Standard Error of Approximation CI: Confidence Interval

**Table 6. Loading matrix related to the exploratory bifactor solution.**

| Items | General factor | Active learning | Collaboration | Learning diversity | High expectation |
|---|---|---|---|---|---|
| 1 | **.570** | **.508** | -.021 | -.097 | -.056 |
| 2 | **.579** | **.449** | -.155 | -.139 | -.195 |
| 3 | **.623** | **.486** | -.177 | -.155 | -.187 |
| 4 | **.466** | **.317** | -.038 | .052 | **.323** |
| 5 | **.516** | **.278** | .100 | -.071 | **.308** |
| 6 | **.356** | **.281** | -.010 | -.017 | .249 |
| 7 | **.427** | **.568** | -.015 | .058 | **.297** |
| 8 | **.485** | **.640** | .029 | -.018 | .145 |
| 9 | **.421** | **.406** | .022 | -.044 | **.391** |
| 10 | **.590** | **.305** | -.068 | .105 | .171 |
| 11 | **.606** | .050 | **.476** | .081 | .082 |
| 12 | **.692** | -.010 | **.704** | .012 | -.085 |
| 13 | **.640** | -.013 | -.022 | **.759** | .058 |
| 14 | **.670** | .019 | -.079 | **.400** | .113 |
| 15 | **.634** | -.154 | -.083 | -.041 | **.493** |
| 16 | **.705** | -.101 | -.063 | -.110 | **.594** |

Loading values larger than .250 are printed in bold.

**Table 7. Internal consistency coefficient (Cronbach's alpha and Omega) for the Educational Practices Questionnaire (EPQ).**

| Item contents summarized | Cronbach's alpha | | Omega (ω) | |
|---|---|---|---|---|
| | Assess perceptions of educational best practices' | | Assess perceptions of educational best practices' | |
| | Presence | Importance in simulation | Presence | Importance in simulation |
| **Active learning** | .860 | .891 | .896 | .930 |
| **Collaboration** | .762 | .832 | .865 | .908 |
| **Learning diversity** | .774 | .832 | .863 | .913 |
| **High expectation** | .769 | .836 | .849 | .911 |
| Total questionnaire | .894 | .915 | .922 | .945 |

questionnaire and for each of the dimensions [35]. The highest alpha value was found for dimension D1. Active Learning. For the rest of the dimensions (D2. Collaboration, D3 Learning diversity and D4. High expectation) the alpha varied between 0.762 and 0.836. Since several dimensions (D2, D3 and D4) only have two items, the Omega index was also calculated. Internal consistency according to McDonald (2013) was adequate ($\omega \geq 0.85$). This instrument has been translated into different languages and countries (Turkish and Portuguese) and reported values similar to those found in our study [27, 45, 46].

The CFA revealed an adequate adjustment of the 4-factor structure consistent with the original version [16].

In our study, a confirmatory factorial analysis was carried out using the generalized least squares method in order to determine whether the scores reproduced the four-dimensional structure on which the original questionnaire is based. The confirmatory factorial analysis showed that all the items presented an adequate factorial load. With respect to the fit indices analyzed for the model, both the absolute fit indices: GFI, RMSR, RMSEA, and the incremental fit indices: AGFI, BBNFI, BBNNFI, CFI and the parsimony indices such as the normed Chi-square all present an acceptable fit. The fit of the model was adequate in relation to the study by Franklin et al. (2014) [27]. In addition, we computed an exploratory factor analysis and observed that the unidimensional factor solution is also acceptable for both scales (*Presence* and *Importance in Simulation*) of the test. In the case of the Presence scale, it is interesting that a bifactor model can be fitted: this means that, while the scale seems to be essentially unidimensional, the four group factors still play a substantial role in the factor model. Our outcomes reinforce the idea that participants' scores can be computed and interpreted for the general factor, but also for the four subscales.

## Limitations

Our study has several limitations. First of all, we selected a sample of convenience from a single university in Barcelona, and therefore, it is possible that our results cannot be generalized to all nursing students. However, the socio-demographic and work characteristics of the students in this study are similar to other universities in Spain and Europe.

Secondly, there is response bias. In other words, the power of the facilitator over the nursing student may also have an impact on the response. This bias has been minimized by conducting the questionnaire anonymously and additionally none of the investigators participated in the simulation activity.

Finally, future studies should investigate the predictive capacity (sensitivity and specificity) of the EPQ-Sp questionnaire, as well as its temporal stability.

## Conclusions

The Educational Practices Questionnaire is a simple and easy-to-administer tool to measure the perception of nursing degree students of the presence of best educational practices and the importance of best practices integrated into clinical simulation. The statistical techniques used in this study enable the addition of solid evidence to support the use of the EPQ questionnaire in Spanish and ensure that simulation judgments are reliable and valid.

## Supporting information

**S1 Data.**
(XLS)

## Acknowledgments

We would like to thank all of the nursing students who participated in the study.

## Author Contributions

**Conceptualization:** Mariona Farrés-Tarafa, Juan Roldán-Merino.

**Data curation:** Mariona Farrés-Tarafa, Barbara Hurtado-Pardos.

**Formal analysis:** Mariona Farrés-Tarafa, Juan Roldán-Merino, Urbano Lorenzo-Seva.

**Investigation:** Mariona Farrés-Tarafa, Juan Roldán-Merino, Barbara Hurtado-Pardos, Ainoa Biurrun-Garrido, Lorena Molina-Raya, Maria-Jose Morera-Pomarede, David Bande, Marta Raurell-Torredà, Irma Casas.

**Methodology:** Mariona Farrés-Tarafa, Juan Roldán-Merino, Ainoa Biurrun-Garrido, Lorena Molina-Raya, Maria-Jose Morera-Pomarede, David Bande, Marta Raurell-Torredà, Irma Casas.

**Resources:** Juan Roldán-Merino, Barbara Hurtado-Pardos, Ainoa Biurrun-Garrido, Lorena Molina-Raya, Maria-Jose Morera-Pomarede, Marta Raurell-Torredà, Irma Casas.

**Software:** Juan Roldán-Merino.

**Supervision:** Mariona Farrés-Tarafa.

**Validation:** Juan Roldán-Merino.

**Visualization:** Mariona Farrés-Tarafa.

**Writing – original draft:** Mariona Farrés-Tarafa, Juan Roldán-Merino.

**Writing – review & editing:** Mariona Farrés-Tarafa, Juan Roldán-Merino, Urbano Lorenzo-Seva, Barbara Hurtado-Pardos, Ainoa Biurrun-Garrido, Lorena Molina-Raya, Maria-Jose Morera-Pomarede, David Bande, Marta Raurell-Torredà, Irma Casas.

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
