## [Decision Letter · Decision Letter 0]

7 Jul 2020

PONE-D-20-16146

Reliability and validity study of the Spanish adaptation of the “Educational Practices Questionnaire” (EPQ).

PLOS ONE

Dear Dr. Merino,

Thank you for submitting your manuscript to PLOS ONE. After careful consideration, we feel that it has merit but does not fully meet PLOS ONE’s publication criteria as it currently stands. Therefore, we invite you to submit a revised version of the manuscript that addresses the points raised during the review process.

While all reviewers saw merit in the manuscript, a minor issues were raised. The manuscript needs a careful review of English.

We look forward to receiving your revised manuscript.

Kind regards,

César Leal-Costa, Ph. D

Academic Editor

PLOS ONE

Journal Requirements:

2. Please clarify in your Methods section whether the questionnaire is published under a CC-BY license, or whether you obtained permission from the publisher to reproduce the questionnaire in this manuscript. Please explain any copyright or restrictions on this questionnaire.

3. Please amend your current ethics statement to address the following concerns:  

a) Did participants provide their written or verbal informed consent to participate in this study?

Reviewers' comments:

Reviewer's Responses to Questions

**Comments to the Author**

1. Is the manuscript technically sound, and do the data support the conclusions?

Reviewer #1: Yes

Reviewer #2: Yes

Reviewer #3: Yes

2. Has the statistical analysis been performed appropriately and rigorously? 

Reviewer #1: Yes

Reviewer #2: Yes

Reviewer #3: Yes

3. Have the authors made all data underlying the findings in their manuscript fully available?

Reviewer #1: Yes

Reviewer #2: Yes

Reviewer #3: Yes

4. Is the manuscript presented in an intelligible fashion and written in standard English?

Reviewer #1: Yes

Reviewer #2: Yes

Reviewer #3: Yes

5. Review Comments to the Author

Reviewer #1: The article covers the following technical aspects:

1.- The study presents the results of primary scientific research.

2.- Results reported have not been published elsewhere.

3.- Experiments, statistics, and other analyses are performed to a high technical standard and are described in sufficient detail.

4.- Conclusions are presented in an appropriate fashion and are supported by the data.

5.- The article is presented in an intelligible fashion and is written in standard English.

6.- The research meets all applicable standards for the ethics of experimentation and research integrity.

7.- The article adheres to appropriate reporting guidelines and community standards for data availability.

The article satisfies the requirements of PlosOne magazine. On the other hand, there are some issues that require clarification from the authors, as well as some suggestions to improve the article.

Abstract

The summary is concrete and adequate. As a suggestion, if the authors consider it appropriate, the results of Cronbach's Alpha and Omega index could be added.

It is exposed that in Spain it is necessary to have validated rubrics that can show the effects of simulation. Is this only in Spain or is it also in other countries?. I propose something similar to this: "As in other countries, in Spain..."

Introduction

This section is correct. It puts the manuscript in context, includes a brief review of the key literature and defines the problem and the need.

I think the aim of the study is not precise enough. The authors’ purpose was “... to translate into Spanish and analyse the reliability and validity of the Educational Practices Questionnaire”. The first part: "Translating into Spanish", the methodology used by the authors has not only been to translate the Educational Practices Questionnaire (EPQ), but also the transcultural adaptation. In fact, both the methodology and Figure 1 describe the realization of the cross-cultural adaptation of the EPQ. I propose to modify it to: "translation and cultural adaptation".

Materials and Methods

Both the population and the methodology used are well justified. Some issues are proposed for the authors' consideration:

In the paragraph: “Others variables:… academic year”. I do not understand why this variable is included, since previously in the “Participants and setting” section, it is stated that the study is carried out from the 2018-2019 academic year. Perhaps you're referring to the nursing course does each student belong to? (1st, 2nd, 3rd, 4th year of the nursing degree)

The pilot test realization time is not well explained in the text, it is only described in the Figure 1. I suggest adding "finally a pilot test was done" in the Procedure section.

Statistical analysis

In the first paragraph, the term CFA models appears, but has not been previously described. A description of this term is given in the next paragraph.

“… Analyse the validity of the construct (CFA)…”

I recommend that you review this aspect and describe it before using it.

In the paragraph: “The following were considered acceptable values: a Crombach´s Alpha of between 0,70 and 0.9 …”. I do not understand why you put this range. If the value is greater than 0.9, how is it considered?

Discussion

In relation to the Fit Indices, the acronyms are set out in the methodology. I believe that to detail them again in the discussion is to fill in the article without providing more content.

References

It is important to check the format of the references. PLOS uses the reference style outlined by the International Committee of Medical Journal Editors (ICMJE), (Vancouver style). References such as 3, 6, 7, 10, 16, 17, 25, 29, 33, 35, 39 are incorrect. Quotations 4 and 5 are incomplete.

In the main text, reference numbers should appear in square brackets [] and not in parentheses. Also, there are some small errors in citations. Example: in paragraph 1, section “Variables and resource of information”. “…integrated in clinical simulation (Jeffries and Rizzolo, 2006).” is in the APA format style.

Figures

In order for the figure to be better understood by itself, it would be good to explain the acronyms that appear on the figures.

Example: figure 1. T1, T2, RTV, etc.

Tables

Tables are self-explanatory. In table 1 the description of the confidence interval is missing at the bottom of the table.

In the methodology reference is made to "Other Variables", I cannot find in the text or in any table the description of the variables: age, sex, teaching shift, academic year, whether they were working, work shift and whether they had previous work experience in the health field. It would be good to make a table with the results obtained.

Reviewer #2: Thank you for giving me the opportunity to review this article. The manuscript is well written and relevant in an educational context (Spanish) where clinical simulation is booming. Congratulations for the excellent work done in adapting the questionnaire to the Spanish context.

Introduction:

As the authors say, it is necessary to have tools to evaluate the effectiveness of the sessions from the point of view of students and facilitators. In your work you carry out an adaptation of a scale on the perception of the participants about the simulation. You say that in Spain there are no validated instruments to evaluate simulated practices, however, this aspect is not completely correct.

There are various scales that have been used in the Spanish educational context to assess student satisfaction with simulation, the assessment of debriefing-DASH- and even the acquisition of skills. Some examples that you can cite in the introduction and/or in the discussion section are the following:

1. Linguistic Validation of the Debriefing Assessment for Simulation in Healthcare in Spanish and Cultural Validation for 8 Spanish Speaking Countries (Muller-Botti et al., 2020)

2. Clinical Simulation as a Learning Tool in Undergraduate Nursing: Validation of a Questionnaire (Alconero-Camarero et al., 2016)

3. To assess the competences of the students, scales such as Clinical Simulation in Nursing Assessment Questionnaire (CLISINAQ) and the Knowledge Management Scale (KMS) have been used in Spain (Díaz Agea, Megías Nicolás, et al., 2019; Sánchez Expósito et al., 2018)

4. To evaluate satisfaction with a simulation methodology, a specific questionnaire was designed for that method (Díaz Agea, Ramos-Morcillo, et al., 2019)

I recommend removing the bold highlighting from the text.

Methods and results.

In the analysis of the internal structure, the authors first present the CFA with the original structure of the scale (4 oblique dimensions). Subsequently, they perform an EFE to analyze the one-dimensional and two-dimensional structure. This is very confusing. It is normally recommended to make sequential use of both types of analysis, whenever the sample size allows it. It is a matter of dividing the sample randomly into two subsamples and exploring the factorial structure underlying the items in the first sample (with an exploratory factor analysis), and then trying to confirm that structure in the other half of the sample, this time by confirmatory factor analysis. The authors must justify why they have carried out the internal structure analyzes first by carrying out a CFA with the original structure of the questionnaire (4 oblique dimensions), and why they then carry out an EFE to analyze the one-dimensional and two-dimensional structure of the questionnaire.

On the other hand, the authors comment that the fit of the model with the original structure of the scale (4 oblique dimensions) in the CFA was good. However, there are fit indices <0.90 and the RMSEA> 0.06. The authors must justify these results or indicate that the model fit was acceptable and not good.

References

Alconero-Camarero, A. R., -Romero, A. G., Sarabia-Cobo, C. M., & Arce, A. M.-. (2016). Clinical simulation as a learning tool in undergraduate nursing: Validation of a questionnaire. Nurse Education Today, 39, 128-134. https://doi.org/10.1016/j.nedt.2016.01.027

Díaz Agea, J. L., Megías Nicolás, A., García Méndez, J. A., Adánez Martínez, M. de G., & Leal Costa, C. (2019). Improving simulation performance through Self-Learning Methodology in Simulated Environments (MAES©). Nurse Education Today, 76, 62-67. https://doi.org/10.1016/j.nedt.2019.01.020

Díaz Agea, J. L., Ramos-Morcillo, A. J., Amo Setien, F. J., Ruzafa-Martínez, M., Hueso-Montoro, C., & Leal-Costa, C. (2019). Perceptions about the Self-Learning Methodology in Simulated Environments in Nursing Students: A Mixed Study. International Journal of Environmental Research and Public Health, 16(23), 4646. https://doi.org/10.3390/ijerph16234646

Muller-Botti, S., Maestre, J. M., del Moral, I., Fey, M., & Simon, R. (2020). Linguistic Validation of the Debriefing Assessment for Simulation in Healthcare in Spanish and Cultural Validation for 8 Spanish Speaking Countries. Simulation in Healthcare, Publish Ahead of Print. https://doi.org/10.1097/SIH.0000000000000468

Sánchez Expósito, J., Leal Costa, C., Díaz Agea, J. L., Carrillo Izquierdo, M. D., & Jiménez Rodríguez, D. (2018). Ensuring relational competency in critical care: Importance of nursing students’ communication skills. Intensive and Critical Care Nursing, 44, 85-91. https://doi.org/10.1016/j.iccn.2017.08.010

Reviewer #3: 1. The study presents the results of original research. Yes

2. Results reported have not been published elsewhere. No

3. Experiments, statistics, and other analyses are performed to a high technical standard and are described in sufficient detail.

Yes, the statistical analyzes are robust, detailed, and very well-grounded.

But, consider this - always use the same rule to report numbers ex: 0, 43, or 0,425 (tenths or hundredths) .

4. Conclusions are presented in an appropriate fashion and are supported by the data. Yes, conclusions are robust and consistent with the results and presented in an appropriate fashion way.

5. The article is presented in an intelligible fashion and is written in standard English.

Yes, the article is presented in a clear, rigorous, and easy to read. But the manuscript needs a careful review of English (USA).

6. The research meets all applicable standards for the ethics of experimentation and research integrity.

Yes.

7. The article adheres to appropriate reporting guidelines and community standards for data availability. The article follows the appropriate reporting guidelines and community standards for data availability.

Yes, but the titles and graphics of the tables can be improved.

The manuscript has great relevance for the teaching practice in nursing. Objectives and methodology are consistent with the object of study and the study complies with all formal and ethical standard requirements.

The sample is robust and adequate, and the treatment and statistical analysis of the data is also robust, detailed and very well based on recent evidence, so it allows conclusions to be drawn, also robust and secure.

But the manuscript needs a careful review of English (USA).

6. PLOS authors have the option to publish the peer review history of their article (what does this mean?). If published, this will include your full peer review and any attached files.

Reviewer #1: No

Reviewer #2: No

Reviewer #3: No

---

## [Author Response · Author response to Decision Letter 0]

1 Aug 2020

RESPONSE TO REVIEWERS

Reference: PONE-D-20-16146

Title: Reliability and validity study of the Spanish adaptation of the “Educational Practices Questionnaire” (EPQ).

Journal Requirements:

We have revised the style requirements of the journal and we have made adjustments.

2. Please clarify in your Methods section whether the questionnaire is published under a CC-BY license, or whether you obtained permission from the publisher to reproduce the questionnaire in this manuscript. Please explain any copyright or restrictions on this questionnaire.

We are not able to publish the scale as the copyright belongs to NLM. However, we have full permission to carry out the study and publish the results and metrics of the items. We have added a table detailing the translation of the items from English to Spanish. For ethical reasons, we have included a section explaining these aspects. We also enclose the formal authorization by the authors of the original tool for the validation in Spanish as a complementary file.

3. Please amend your current ethics statement to address the following concerns: 

a) Did participants provide their written or verbal informed consent to participate in this study?.

We thank the reviewer for this precision. We obtained written informed consent from each participant and we have modified the manuscript accordingly.

We apologize for this oversight and we have uploaded the anonymized data set that will enable others to replicate our findings as per the instructions by the journal.

 RESPONSES TO REVIEWER 1

General comments: The study addresses an important subject that was studied using an appropriate design but the number with outcomes was small and follow up was short. The authors did well to recommend future studies with longer follow up but should also add big sample size.

Abstract

The summary is concrete and adequate. As a suggestion, if the authors consider it appropriate, the results of Cronbach's Alpha and Omega index could be added.

Thank you for these comments. The authors have considered adding the results of Cronbach’s Alpha and Omega index, but there are a lot of values and we have decided that it would not be appropriate /would not add sufficient value to warrant it.

It is exposed that in Spain it is necessary to have validated rubrics that can show the effects of simulation. Is this only in Spain or is it also in other countries?. I propose something similar to this: "As in other countries, in Spain..."

We thank the reviewer for this point and have amended the manuscript accordingly.

Introduction

This section is correct. It puts the manuscript in context, includes a brief review of the key literature and defines the problem and the need.

We thank the reviewer for these comments.

I think the aim of the study is not precise enough. The authors’ purpose was “... to translate into Spanish and analyse the reliability and validity of the Educational Practices Questionnaire”. The first part: "Translating into Spanish", the methodology used by the authors has not only been to translate the Educational Practices Questionnaire (EPQ), but also the transcultural adaptation. In fact, both the methodology and Figure 1 describe the realization of the cross-cultural adaptation of the EPQ. I propose to modify it to: "translation and cultural adaptation".

We thank the reviewer for this useful indication about the cultural dimension. We have adapted the text accordingly. 

Materials and Methods

Both the population and the methodology used are well justified. Some issues are proposed for the authors' consideration:

In the paragraph: “Others variables:… academic year”. I do not understand why this variable is included, since previously in the “Participants and setting” section, it is stated that the study is carried out from the 2018-2019 academic year. Perhaps you're referring to the nursing course does each student belong to? (1st, 2nd, 3rd, 4th year of the nursing degree)

The authors accept this point that this variable could lead to errors and we have eliminated it from the text.

The pilot test realization time is not well explained in the text, it is only described in the Figure 1. I suggest adding "finally a pilot test was done" in the Procedure section.

We thank the reviewer for pointing this out; we have now included the pilot test in the text to make this part clearer.

Statistical analysis

In the first paragraph, the term CFA models appears, but has not been previously described. A description of this term is given in the next paragraph.

“… Analyse the validity of the construct (CFA)…”

I recommend that you review this aspect and describe it before using it.

We thank the reviewer for pointing out this oversight and we have corrected it.

In the paragraph: “The following were considered acceptable values: a Crombach´s Alpha of between 0,70 and 0.9 …”. I do not understand why you put this range. If the value is greater than 0.9, how is it considered?

We take the reviewer’s point and we have rewritten this paragraph. We believe it is now clearer.

Discussion

In relation to the Fit Indices, the acronyms are set out in the methodology. I believe that to detail them again in the discussion is to fill in the article without providing more content.

We have eliminated the description of the acronyms from the discussion.

In the main text, reference numbers should appear in square brackets [] and not in parentheses. Also, there are some small errors in citations. Example: in paragraph 1, section “Variables and resource of information”. “…integrated in clinical simulation (Jeffries and Rizzolo, 2006).” is in the APA format style.

We apologize for this omission and we have checked all the references and corrected the errors.

Figures

In order for the figure to be better understood by itself, it would be good to explain the acronyms that appear on the figures.

Example: figure 1. T1, T2, RTV, etc.

We have added the description of the acronyms to the figures to improve clarity.

Tables

Tables are self-explanatory. In table 1 the description of the confidence interval is missing at the bottom of the table.

We apologize for this omission and we have added the description of the confidence interval (Ci).

In the methodology reference is made to "Other Variables", I cannot find in the text or in any table the description of the variables: age, sex, teaching shift, academic year, whether they were working, work shift and whether they had previous work experience in the health field. It would be good to make a table with the results obtained.

We have included a table with the descriptions of the variables as recommended by the reviewer.

RESPONSES TO REVIEWER 2

Introduction:

As the authors say, it is necessary to have tools to evaluate the effectiveness of the sessions from the point of view of students and facilitators. In your work you carry out an adaptation of a scale on the perception of the participants about the simulation. You say that in Spain there are no validated instruments to evaluate simulated practices, however, this aspect is not completely correct.

There are various scales that have been used in the Spanish educational context to assess student satisfaction with simulation, the assessment of debriefing-DASH- and even the acquisition of skills. Some examples that you can cite in the introduction and/or in the discussion section are the following:

1. Linguistic Validation of the Debriefing Assessment for Simulation in Healthcare in Spanish and Cultural Validation for 8 Spanish Speaking Countries (Muller-Botti et al., 2020)

2. Clinical Simulation as a Learning Tool in Undergraduate Nursing: Validation of a Questionnaire (Alconero-Camarero et al., 2016)

3. To assess the competences of the students, scales such as Clinical Simulation in Nursing Assessment Questionnaire (CLISINAQ) and the Knowledge Management Scale (KMS) have been used in Spain (Díaz Agea, Megías Nicolás, et al., 2019; Sánchez Expósito et al., 2018)

4. To evaluate satisfaction with a simulation methodology, a specific questionnaire was designed for that method (Díaz Agea, Ramos-Morcillo, et al., 2019) studies with larger sample size

We thank the reviewer for this excellent summary of available scales and we have edited the introduction to include the references mentioned.

Methods and results.

In the analysis of the internal structure, the authors first present the CFA with the original structure of the scale (4 oblique dimensions). Subsequently, they perform an EFE to analyze the one-dimensional and two-dimensional structure. This is very confusing. It is normally recommended to make sequential use of both types of analysis, whenever the sample size allows it. It is a matter of dividing the sample randomly into two subsamples and exploring the factorial structure underlying the items in the first sample (with an exploratory factor analysis), and then trying to confirm that structure in the other half of the sample, this time by confirmatory factor analysis. The authors must justify why they have carried out the internal structure analyzes first by carrying out a CFA with the original structure of the questionnaire (4 oblique dimensions), and why they then carry out an EFE to analyze the one-dimensional and two-dimensional structure of the questionnaire.

We agree with the reviewer that the usual procedure when first analyzing a questionnaire is: (1) to randomly divide the sample in two halves; and (2) to compute an EFA in the first subsample, and a CFA in the second subsample. However, this is the procedure when researchers do not have a hypothesis about the factor structure underlying the questionnaire, which his was not the case here. As the questionnaire has already been analyzed extracting four factors (with a well-defined set of items defining each factor), our hypothesis was that in the Spanish population the same factor model would be a valid one. This is the reason why we computed a CFA first in order to test the expected hypothesis. We could not accept that the four-factor model was valid for the Spanish population.

Because of this we needed to abandon our four factor-hypothesis and to explore our sample in order to decide which model should be expected in the Spanish population. In order to do so, we computed an EFA, and we concluded that the most appropriate factor model is a mixture of unidimensional model (a single general factor), and a two-factor highly correlated model. This is the reason why we explored our data in the context of a bifactor model (that combines both, a single general factor and two group factors).

On the other hand, the authors comment that the fit of the model with the original structure of the scale (4 oblique dimensions) in the CFA was good. However, there are fit indices <0.90 and the RMSEA> 0.06. The authors must justify these results or indicate that the model fit was acceptable and not good.

We thank the reviewer for this observation. We have made the correction and adapted our text to take this into account.

RESPONSES TO REVIEWER 3

Experiments, statistics, and other analyses are performed to a high technical standard and are described in sufficient detail.

Yes, the statistical analyzes are robust, detailed, and very well-grounded.

But, consider this - always use the same rule to report numbers ex: 0, 43, or 0,425 (tenths or hundredths).

We thank the reviewer for this comment and we have taken it into account in our results section.

The article is presented in an intelligible fashion and is written in standard English.

Yes, the article is presented in a clear, rigorous, and easy to read. But the manuscript needs a careful review of English (USA).

We thank the reviewer for these comments. We have had the manuscript edited by an English speaking editor and we hope the manuscript is now more uniform regarding the use of English.

The article adheres to appropriate reporting guidelines and community standards for data availability. The article follows the appropriate reporting guidelines and community standards for data availability.

Yes, but the titles and graphics of the tables can be improved.

We accept these comments. We have made adjustments according to suggestions of all three reviewers and we think that the tables have now been improved.

---

## [Decision Letter · Decision Letter 1]

25 Aug 2020

PONE-D-20-16146R1

Reliability and validity study of the Spanish adaptation of the “Educational Practices Questionnaire” (EPQ).

PLOS ONE

Dear Dr. Merino,

Thank you for submitting your manuscript to PLOS ONE. After careful consideration, we feel that it has merit but does not fully meet PLOS ONE’s publication criteria as it currently stands. Therefore, we invite you to submit a revised version of the manuscript that addresses the points raised during the review process.

We look forward to receiving your revised manuscript.

Kind regards,

César Leal-Costa, Ph. D

Academic Editor

PLOS ONE

Reviewers' comments:

Reviewer's Responses to Questions

**Comments to the Author**

1. If the authors have adequately addressed your comments raised in a previous round of review and you feel that this manuscript is now acceptable for publication, you may indicate that here to bypass the “Comments to the Author” section, enter your conflict of interest statement in the “Confidential to Editor” section, and submit your "Accept" recommendation.

Reviewer #1: All comments have been addressed

Reviewer #2: All comments have been addressed

Reviewer #3: All comments have been addressed

2. Is the manuscript technically sound, and do the data support the conclusions?

Reviewer #1: Yes

Reviewer #2: Yes

Reviewer #3: Yes

3. Has the statistical analysis been performed appropriately and rigorously? 

Reviewer #1: Yes

Reviewer #2: Yes

Reviewer #3: Yes

4. Have the authors made all data underlying the findings in their manuscript fully available?

Reviewer #1: Yes

Reviewer #2: Yes

Reviewer #3: Yes

5. Is the manuscript presented in an intelligible fashion and written in standard English?

Reviewer #1: Yes

Reviewer #2: Yes

Reviewer #3: Yes

6. Review Comments to the Author

Reviewer #1: Dear authors,

The article has improved considerably. The authors have made the changes eloquently. There are only 2 details more to be considered:

1.- In my opinion, in the Abstract the objective is incomplete in relation to the general work. (translation and cultural adaptation).

2.- There is a lack of homogenization in the pagination of some references. Most of them show the complete initial and final number, but the references: 1, 14, 15, 40, 43, 44, 45. They show an abbreviated model.

Thank you very much for your efforts.

kind regards.

Reviewer #2: The authors have adequately responded to the reviewer's concerns. The changes proposed by the reviewer have been partially complied with.

Reviewer #3: The authors improved the manuscript according to the three reviewers' recommendations, so the manuscript can now be accepted as is.

Also, the manuscript presents a robust validation study of a relevant instrument to support nurse teaching practice in Spain.

7. PLOS authors have the option to publish the peer review history of their article (what does this mean?). If published, this will include your full peer review and any attached files.

Reviewer #1: No

Reviewer #2: No

Reviewer #3: No

---

## [Author Response · Author response to Decision Letter 1]

26 Aug 2020

6. Review Comments to the Author

Reviewer #1: Dear authors,

The article has improved considerably. The authors have made the changes eloquently. There are only 2 details more to be considered:

1.- In my opinion, in the Abstract the objective is incomplete in relation to the general work. (translation and cultural adaptation).

We thank the reviewer for this useful indication. We have adapted the text accordingly.

2.- There is a lack of homogenization in the pagination of some references. Most of them show the complete initial and final number, but the references: 1, 14, 15, 40, 43, 44, 45. They show an abbreviated model.

We apologize for this omission and we have checked all the references.

---

## [Editor Report · Decision Letter 2]

28 Aug 2020

Reliability and validity study of the Spanish adaptation of the “Educational Practices Questionnaire” (EPQ).

PONE-D-20-16146R2

Dear Dr. Merino,

We’re pleased to inform you that your manuscript has been judged scientifically suitable for publication and will be formally accepted for publication once it meets all outstanding technical requirements.

Kind regards,

César Leal-Costa, Ph. D

Academic Editor

PLOS ONE
---

## [Editor Report · Acceptance letter]

3 Sep 2020

PONE-D-20-16146R2 

Reliability and validity study of the Spanish adaptation of the “Educational Practices Questionnaire” (EPQ). 

Dear Dr. Roldán-Merino:

I'm pleased to inform you that your manuscript has been deemed suitable for publication in PLOS ONE. Congratulations! Your manuscript is now with our production department. 

Kind regards, 

on behalf of

Dr. César Leal-Costa 

Academic Editor

PLOS ONE